# Texture Evolution by Strain-Induced Boundary Migration during Hot Deformation of Fe-3.0 wt.% Si Alloy: Experiment and Modeling

Guangshuai Shao [1], Xi Chen [1], Yuhui Sha [1,*], Fang Zhang [1], Zhenghua He [1,2] and Liang Zuo [1]

[1] Key Laboratory for Anisotropy and Texture of Materials, Ministry of Education, Northeastern University, Shenyang 110819, China; guangshuaishao@126.com (G.S.); chenxineu@163.com (X.C.); zhangf@smm.neu.edu.cn (F.Z.); hezhenghua@mail.neu.edu.cn (Z.H.); lzuo@mail.neu.edu.cn (L.Z.)
[2] School of Materials Science and Engineering, Shenyang University of Technology, Shenyang 110870, China
* Correspondence: yhsha@mail.neu.edu.cn; Tel.: +86-24-8369-1569

**Abstract:** Texture and microstructure evolution during high-temperature plane-strain compression in Fe-3.0 wt.% Si alloy has been investigated by micro-texture analysis and modeling. In this study, hot deformation test is performed on the temperature range of 900 °C~1150 °C with a strain rate scope of 0.01 s$^{-1}$~5 s$^{-1}$, and the effect of deformation parameters is investigated by means of electron backscattered diffraction. Nucleation and growth assisted by strain-induced boundary migration result in strong {001}<110> and {001}<210> texture components with low Taylor factors, and the grain size of λ fiber increases significantly by consuming the {111}<110> and {111}<112> texture components with high Taylor factors. The critical Taylor factor above which nucleation by strain-induced boundary migration cannot occur, decreases continuously during hot deformation. With the decreasing critical Taylor factor, the increment rate of low-Taylor-factor orientation depends more sensitively on Taylor factor than the decrement rate of high-Taylor-factor orientation. The boundary separating enhanced and weakened orientations moves towards lower Taylor factor with the deformation proceeding, and medium-Taylor-factor texture components may experience a reversed change from enhancement to weakness. A quantitative model is proposed to describe texture development by incorporating the oriented nucleation probability dependent on a variable critical Taylor factor and the selective growth driven by a variable Taylor factor difference between adjacent grains. The present work can provide an efficient method for optimizing hot deformation texture by means of strain-induced boundary migration.

**Keywords:** texture; strain-induced boundary migration (SIBM); hot deformation; Taylor factor; silicon steel

## 1. Introduction

Hot deformation texture plays a vital role in texture development through thermo-mechanical processing. The textures in cold deformation and annealing are known to be inherited from hot deformation texture [1,2]. For example, recrystallization γ fiber (<111>//ND) is the ideal product texture for deep drawability in interstitial-free steels, where γ fiber and λ fiber (<001>//ND) are the favorable and unfavorable hot deformation textures, respectively [3,4]. In contrast, λ fiber is the ideal texture for magnetic properties in non-oriented electrical steels, which derives from hot deformation λ fiber and retains during cold deformation and final annealing [5–7]. Therefore, the precise control on hot deformation texture is the prerequisite to optimize the final product texture and performance [8,9].

The texture evolution during hot deformation results from the crystal orientation rotation [10,11], as well as the formation and migration of high-angle grain boundaries [12–14], which is closely related to initial texture and deformation parameters. The difference

in stored strain energy between adjacent deformed grains has a great effect on texture evolution in terms of strain-induced boundary migration (SIBM) [15,16]. The enhancement of texture components with low stored strain energy has been frequently observed under hot deformation [17,18].

Taylor factor is normally used to represent the stored strain energy dependent on grain orientation approximately [19]. The texture components with various Taylor factors can exhibit a variety of evolution during hot deformation. In torsion of Ti-IF steel, {112}<111> with Taylor factor 1.7 increases, and {110}<112> with Taylor factor 2.5 decreases as the true strain changes from 1.9 to 4.9 [20]. In plane-strain compression of Fe-3.0 wt.% Si alloy, {001}<110> with Taylor factor 2.1 increases and $\gamma$ fiber with Taylor factor 3.5 reduces with the strain rate decreasing from $5.0 \times 10^{-3}$ s$^{-1}$ to $5.0 \times 10^{-5}$ s$^{-1}$, whereas {112}<110> having Taylor factor 3.2, increases from $5.0 \times 10^{-3}$ s$^{-1}$ to $5.0 \times 10^{-4}$ s$^{-1}$ and reduces from $5.0 \times 10^{-4}$ s$^{-1}$ to $5.0 \times 10^{-5}$ s$^{-1}$ [21]. In torsion of Ni-30 wt.% Fe alloy, {111}<110> with Taylor factor 1.7 increases through the dynamic recrystallization (DRX) process, while {110}<110> with Taylor factor 3.0 decreases below 90% DRX fraction and remains nearly unchanged above 90% DRX fraction [22]. In plane-strain compression of Ni-30 wt.% Fe-Nb-C alloy, {001}<100> with Taylor factor 2.5 does not change obviously below 20% and increases significantly at a larger DRX fraction, while {110}<112> with Taylor factor 2.7 decreases first and then remains constant beyond 20% DRX fraction [23].

The enhancement of texture components with low Taylor factors is basically related with the boundary migration of original grains or newly formed grains assisted by a strain-induced bulge process. Kestens [24] and Sidor [25–27] proposed a low-Taylor-factor nucleation model that deformation orientations with Taylor factors below a critical value nucleate with a constant probability. Baczynski and Jonas [20] suggested that texture components below critical Taylor factor can only nucleate with the probability dependent on both critical and minimum Taylor factors. Actually, various texture components show distinct evolution kinetics and even reversed tendency with the continuously changing Taylor factor range. However, the texture evolution process during hot deformation assisted by SIBM has not yet well been described in consideration of a changing Taylor factor range, although it is of great importance for in-depth understanding and accurate control of hot deformation texture.

In this paper, hot deformation texture in Fe-3.0 wt.% Si alloy during plane-strain compression at different deformation temperatures and strain rates was investigated. A quantitative model was proposed to accurately describe the texture evolution due to SIBM by tracking the variation of Taylor factor distribution during the hot deformation process.

## 2. Materials and Methods

Fe-3.0 wt.% Si hot rolled plates, containing 0.003 wt.% C, 3.0 wt.% Si, 0.02 wt.% Mn, 0.001 wt.% S, 0.019 wt.% P and balance Fe, were annealed at 1150 °C for 10 min to complete recrystallization. Hot deformation specimens with 5 mm in height (normal direction, ND), 20 mm in width (transverse direction, TD), and 15 mm in longitudinal length (rolling direction, RD) were prepared from the annealed hot rolled plates for plane-strain compression on MMS-200 thermo-simulation machine. The specimen directions were defined in reference to rolling mode, as shown in Figure 1. The temperature was controlled by the electric current through the anvils and measured by the thermocouples welded onto the surface at the thickness center of each specimen. The specimens were first heated at a rate of 10 °C·s$^{-1}$ to the target deformation temperature and held for 1 min to eliminate in-specimen temperature gradient. The deformed specimens were required to have different states. One was the deformed state without SIBM, and the others were deformed states with different progressions of SIBM. Compression testing was carried out under the strain rates of 1, 0.1, and 0.01 s$^{-1}$ at 1150 °C, as well as 5 s$^{-1}$ at 900 °C. After compressed to 50% reduction, the specimens were quenched in water immediately to avoid the occurrence of post-deformation thermally activated phenomena such as static recrystallization or grain growth. Three specimens were used for each deformation condition to ensure statistics.

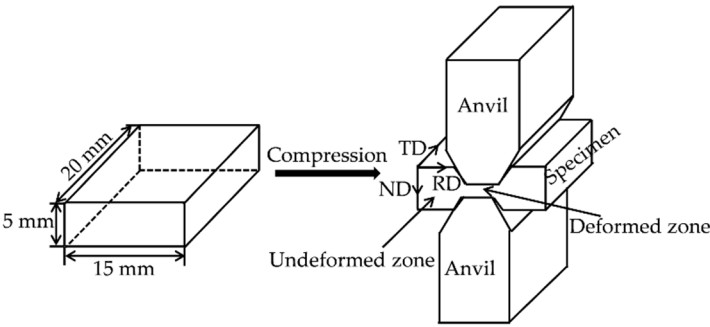

**Figure 1.** Schematic drawing of the plane-strain compression.

Texture and microstructure were analyzed by electron backscattered diffraction (EBSD) on a JEOL JSM-7001F scanning electron microscope with an electron accelerating voltage of 15 KV at a working distance of 15 mm. The undeformed and deformed parts of the compressed specimen were cutting separation, and the mid-plane of RD-TD sections of both parts were prepared for EBSD specimens. EBSD measurement was performed using a step size of 15 μm for general characterization, and a step size of 4 μm was adopted for local enlarged map. To facilitate the EBSD measurement, a scanned area of 5 mm × 20 mm for each specimen was cut into two parts with the area of 5 mm × 10 mm. Thus, a total area of about 15 mm × 20 mm were scanned for each deformation condition. The specimens for EBSD analysis were prepared by first mechanical polishing, and then electropolishing in a solution of 92% ethanol and 8% perchloric acid for 15 s at 20 V at the temperature of 0 °C.

## 3. Results and Discussion

### 3.1. Microstructure Evolution during Hot Deformation

Figure 2 illustrates the orientation image map and texture of hot compression specimens reconstructed from EBSD data. The initial microstructure mainly consists of equiaxed grains distributed between 100~400 μm and composed of γ fiber with peak at {111}<112>, α fiber (<110>//RD) with peak at {114}<110> and λ fiber with peak at {001}<210>.

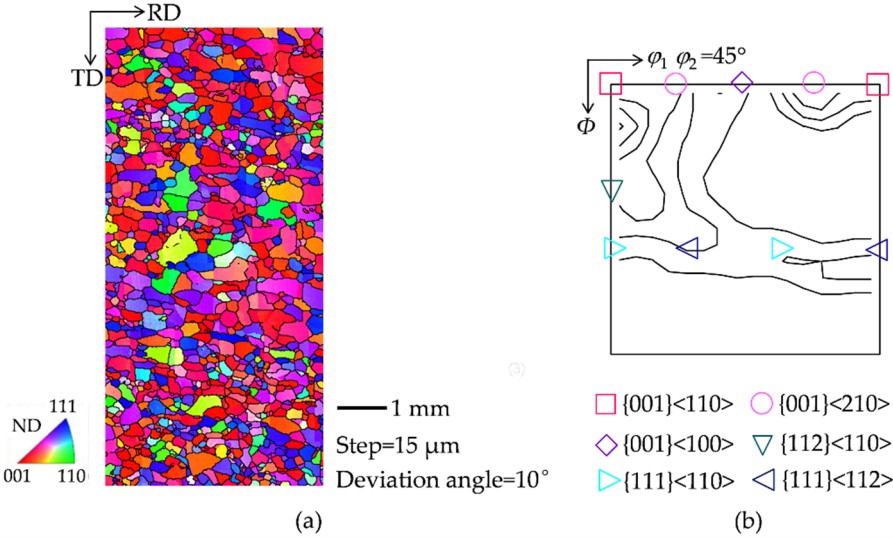

(a)  (b)

**Figure 2.** (**a**) Orientation image map and (**b**) constant $\varphi_2 = 45°$ section of ODF (levels: 1, 2, 3 . . . ) of hot compression specimens.

Figure 3 shows the orientation image maps and average grain size of undeformed and deformed parts of the compressed specimens reconstructed from EBSD data under different hot deformation conditions. The undeformed parts experienced the same heating process as deformed parts but without deformation. After compressed at 900 °C with 5 s$^{-1}$, most

grains are highly elongated with small bulges of original grain boundaries and small grains having low Taylor factors (Figure 3b). There appear some large λ grains with the lowest Taylor factors, indicating the local occurrence of SIBM. When deformed at 1150 °C with 1 s$^{-1}$, most grain boundaries of λ grains are featured with bulges and a lot of small new λ grains form in the deformed grains with relatively high Taylor factors (Figure 3d). A local enlarged map of region I in Figure 3d clearly shows the bulges along the short boundaries of small λ grains (Figure 3i). This suggests that the bulges of grain boundaries and formation of small grains by SIBM dominate the evolution of microstructure during hot deformation. Compressed at 1150 °C with 0.1 s$^{-1}$, the average size of λ grains increases accompanied by the growth of bulges and new λ grains, contributing to the evident enhancement of λ texture (Figure 3f). After deformation at 1150 °C with 0.01 s$^{-1}$, the average size of λ grains increases further and λ texture consumes most of the other textures (Figure 3h). The straight grain boundary dominates the microstructure, and λ grains consume most of deformed grains with high and medium Taylor factors. Figure 3i shows the average grain size of undeformed and deformed grains under different deformation conditions. Based on the comparison of microstructure, average grain size and texture of undeformed parts, the average grain size increases slightly with increasing deformation temperature and time; however, λ grains have no advantage over other textures and there are no bulges and new grains (Figure 3a,c,e,g). Therefore, SIBM is responsible for the microstructure and texture evolution during hot compression in the present study.

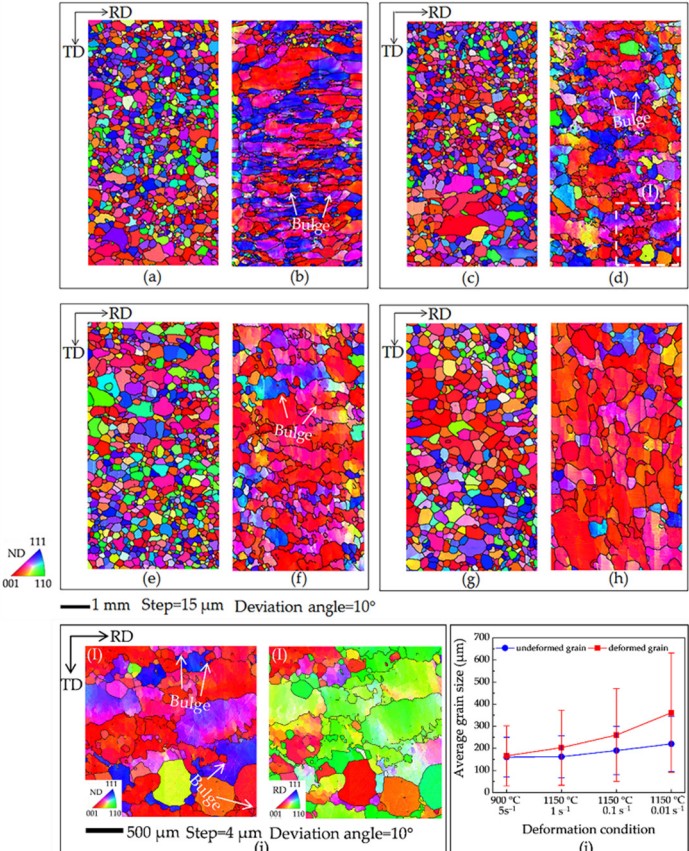

**Figure 3.** Orientation image maps of uncompressed and compressed parts of the specimens under different deformation conditions of 900 °C and 5 s$^{-1}$ (**a,b**), 1150 °C and 1 s$^{-1}$ (**c,d**), 1150 °C and 0.1 s$^{-1}$ (**e,f**), 1150 °C and 0.01 s$^{-1}$ (**g,h**), (**i**) local enlarged map of region I in Figure 3d as well as (**j**) average grain size of undeformed and deformed grains under different deformation conditions.

### 3.2. Texture Evolution during Hot Deformation

Figure 4 presents texture characteristics under different hot deformation conditions. Typical rolling textures consisting of α and γ fibers form after compression at 900 °C with 5 s$^{-1}$ (Figure 4a), corresponding to little bulges and new grains. With the increasing deformation temperature and decreasing strain rate, {001}<110> and {001}<210> components are enhanced continuously. The {112}<110> component exhibits a nonmonotonic change that it decreases by compression with 1150 °C and 1 s$^{-1}$ (Figure 4b) and remains nearly constant after compression with 1150 °C and 0.1 s$^{-1}$ (Figure 4c), while decreases again with further decreasing strain rate (Figure 4d). In contrast, the γ fiber decreases continuously in hot deformation process, and is nearly exhausted after compression with 1150 °C and 1 s$^{-1}$ (Figure 4b). Onuki [21] observed the similar result of the increase of {001}<110> and decrease of γ fiber in plane-strain compression of Fe-3.0 wt.% Si alloy and considered the γ fiber to be consumed by the preferential growth of {001}<110>. Thus, the texture evolution confirms that the hot deformation process is characteristic of low-Taylor-factor texture components consuming those with relatively high Taylor factors.

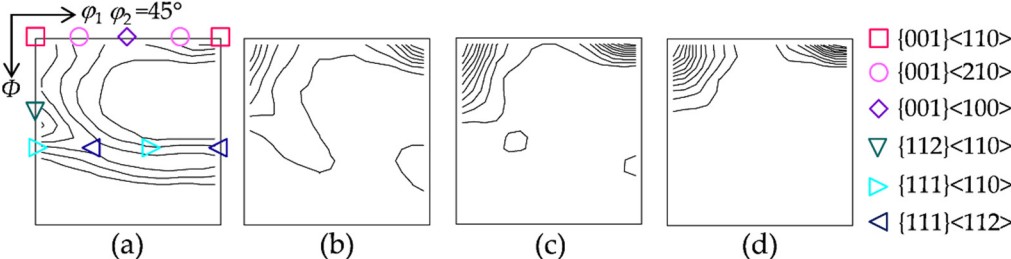

**Figure 4.** Constant $\varphi_2$ = 45° section of ODFs (levels: 1, 2, 3 . . . ) after hot compression with (**a**) 900 °C and 5 s$^{-1}$, (**b**) 1150 °C and 1 s$^{-1}$, (**c**) 1150 °C and 0.1 s$^{-1}$, (**d**) 1150 °C and 0.01 s$^{-1}$.

To identify SIBM effect, the contribution of SIBM to texture evolution is extracted by subtracting the hot deformation texture without SIBM from the hot deformation texture with SIBM. The four hot deformation conditions represent different texture development stages by SIBM, which can be quantitatively described by total low-Taylor-factor volume fraction increments. The deformation with 900 °C and 5 s$^{-1}$ approximately corresponds to the stage with little SIBM effect. Figure 5 shows the orientation density variation of main texture components between adjacent stages. At an early stage, {001}<110> component exhibits the largest increase in orientation density, and {001}<210> and {001}<100> components keep nearly constant, while {112}<110>, {111}<110> and {111}<112> components present a similar decrease. In the second stage, {001}<110> and {001}<210> components have a large increase in orientation density, while {001}<100>, {112}<110>, {111}<110> and {111}<112> components remain nearly unchanged. In the third stage, {001}<110> and {001}<210> components exhibit an increase in orientation density, while {001}<100>, {111}<110> and {111}<112> components keep nearly constant, and {112}<110> component shows a large decrease. Accordingly, at an early stage of the SIBM effect, the critical orientation boundary separating the enhanced and weakened texture components (olive lines in Figure 5) is the orientation line deviated about 30° from γ fiber. Afterwards, there occurs a moderate change in orientation density but with an obvious convergence of critical orientation boundary towards {001}<110>. The orientation density variation and shift of critical orientation boundary indicate that the texture components with high Taylor factors are preferentially consumed by those with low Taylor factors, and the texture components with medium Taylor factors may experience a reversed change from enhancement to weakness with the proceeding SIBM effect.

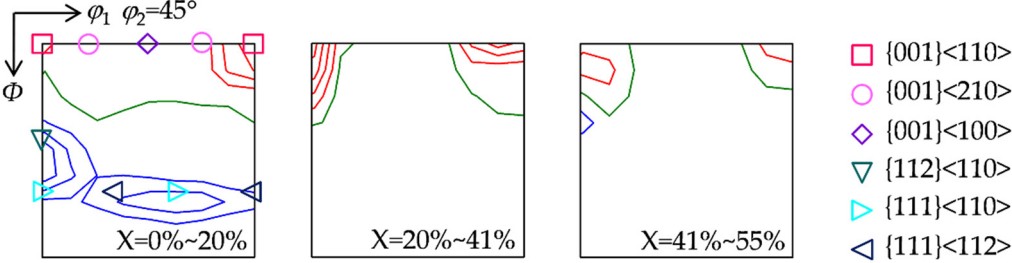

**Figure 5.** Orientation density variation of main texture components between adjacent stages of SIBM effect (red line levels: 1, 2, 3 . . . ; olive line level: 0; blue line levels: −1, −2, −3 . . . ). X indicates total low-Taylor-factor volume fraction increments.

### 3.3. Texture Evolution Model

Baczynski and Jonas [20] proposed a low-Taylor-factor nucleation model, where the nucleation probability ($P_{g_i}^N$) of low-Taylor-orientation $g_i$ is written as follows:

$$P_{g_i}^N = \exp\left[-\left(\frac{M_{g_i} - M_{\min}}{M_0 - M_{\min}}\right)^n\right] \tag{1}$$

Here, $n$ is a Gaussian exponent, $M_0$ is the critical Taylor factor, $M_{\min}$ is the minimum Taylor factor, and $M_{g_i}$ is the Taylor factor of orientation $g_i$. Beladi [18,22] reported the prominent mechanism for texture evolution in Ni-30%Fe austenitic alloy is the preferred nucleation of low-Taylor-factor component by SIBM. In the case of microstructure and texture evolution by SIBM, $P_{g_i}^N$ can actually reflect the fraction of grain boundaries surrounding orientation $g_i$ that can bulge. Based on the shift of critical orientation boundary during hot deformation in the present study, a variable critical Taylor factor is required in the description of texture evolution by SIBM.

Here, a quantitative model is proposed to differentiate the evolution of various texture components, where the Taylor factors involved in nucleation probability (Equation (1)) and growth rate dependent on the Taylor factor difference between adjacent grains are all employed as a variable. The volume fraction increment ($\Delta V_{g_i}$) of a low-Taylor-factor orientation $g_i$ within one strain step is written as:

$$\Delta V_{g_i} = K P_{g_i}^N S_{g_i}\left(\overline{M_{g_i}^A} - M_{g_i}\right) \tag{2}$$

where $K$ is a constant, $S_{g_i}$ is the grain boundary area of orientation $g_i$, $\overline{M_{g_i}^A}$ is the averaged Taylor factor of adjacent grains surrounding the grains with orientation $g_i$. Neglecting the grain size difference among low-Taylor-factor texture components, the proportion of volume fraction increment of orientation $g_i$ ($f_{g_i}$) in total increments of all low-Taylor-factor texture components within one step is:

$$f_{g_i} = P_{g_i}^N V_{g_i}\left(\overline{M_{g_i}^A} - M_{g_i}\right) \Big/ \left(\sum_{i=1}^{l} P_{g_i}^N V_{g_i}\left(\overline{M_{g_i}^A} - M_{g_i}\right)\right) \tag{3}$$

where $l$ is the number of low-Taylor-factor texture components, $V_{g_i}$ is the volume fraction of orientation $g_i$. Similarly, the proportion of volume fraction decrement of orientation $g_j$ ($f_{g_j}$) in total decrements of all high-Taylor-factor texture components within one step is:

$$f_{g_j} = V_{g_j}\left(M_{g_j} - \overline{M_{g_j}^A}\right) \Big/ \left(\sum_{j=1}^{m} V_{g_j}\left(M_{g_j} - \overline{M_{g_j}^A}\right)\right) \tag{4}$$

where $m$ is the number of high-Taylor-factor texture components, $V_{g_j}$ is the volume fraction of orientation $g_j$, $M_{g_j}$ is the Taylor factor of orientation $g_j$, and $\overline{M_{g_j}^A}$ is the averaged Taylor

factor of adjacent grains surrounding the grains with orientation $g_j$. Then, the volume fraction increment of low-Taylor-factor orientation $g_i$ ($\Delta V_{g_i}$) within one step is:

$$\Delta V_{g_i} = \Delta V \, f_{g_i} \tag{5}$$

Here, $\Delta V$ is the total low-Taylor-factor volume fraction increments within one step. Likewise, the volume fraction decrement of high-Taylor-factor orientation $g_j$ ($\Delta V_{g_j}$) within one step is:

$$\Delta V_{g_j} = \Delta V f_{g_j} \tag{6}$$

In Equations (3) and (4), the parameters of $l$, $m$, $V_{g_i}$, $V_{g_j}$, $\overline{M_{g_i}^A}$ and $\overline{M_{g_j}^A}$ vary with dynamic process, which are updated after each step. Taylor factor is calculated by full constraint Taylor model under plane strain, and 48 slip systems ($12 \times \{110\}<111>$, $12 \times \{112\}<111>$ and $24 \times \{123\}<111>$) are considered [28,29]. The $M_0$ value is obtained as the maximum $M_{g_i}$ value among all low-Taylor-factor texture components within one step. Thus, texture evolution based on SIBM can be accurately described by means of the oriented nucleation probability dependent on a variable $M_0$ and the selective growth driven by a variable Taylor factor difference between adjacent grains.

*3.4. Texture Evolution by SIBM*

Texture evolution during hot deformation can be quantitatively deduced based on the proposed model, as shown in Figure 6. The initial texture used in calculation concentrates in α and γ fibers including {001}<110>, {001}<210>, {001}<100>, {112}<110>, {111}<110> and {111}<112> components, which is reconstructed from the measured deformation texture without SIBM (Figure 4a). The calculated texture at total low-Taylor-factor volume fraction increments of 20%, 40%, and 60% are evidently consistent with the measurements corresponding to three deformation conditions at 1150 °C (Figure 4b–d). With the increasing total low-Taylor-factor volume fraction increment, {001}<110> and {001}<210> components increase continuously, {112}<110> component gradually decreases and is exhausted at low-Taylor-factor increment of 40%, while γ fiber reduces to zero at 50% low-Taylor-factor increment. The {001}<100> component remains weak and changes little during texture evolution by SIBM. Figure 7 plots the calculated $M_0$ values and orientation density variation of main texture components as a function of total low-Taylor-factor volume fraction increment. The calculation based on the proposed model agrees well with the EBSD measurement in Figure 5.

Baczynski and Jonas [21] calculated the texture evolution during hot deformation with a quantitative model incorporating the oriented nucleation probability dependent on an estimated constant $M_0$ value and uniform growth. Figure 8 shows the calculated orientation density variation of main texture components with the model of Baczynski and Jonas between adjacent stages. Throughout hot deformation, the orientation density variation of main texture components diverges significantly from the EBSD measurement and the critical orientation boundary of the high/low-Taylor-factors remains unchanged. Therefore, the calculation with an estimated constant $M_0$ value and uniform growth is not sufficiently sensitive to capture the main features of texture evolution. Accordingly, the consideration of the dynamic evolution of $M_0$ value and selective growth in the model is necessary to quantitatively describe texture evolution by SIBM.

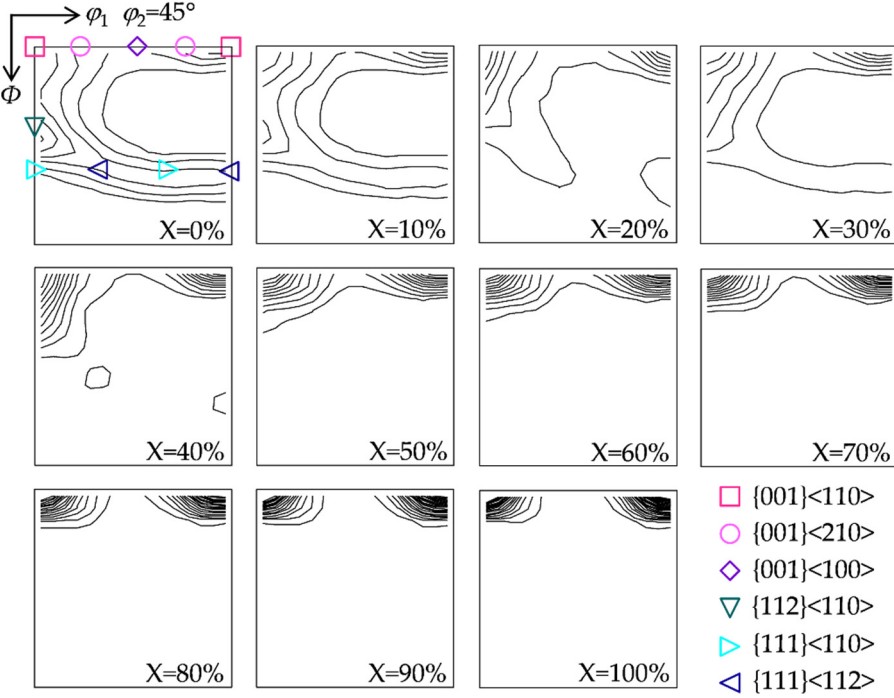

**Figure 6.** The calculated deformation texture (levels: 1, 2, 3 . . . ) at different total low-Taylor-factor volume fraction increments (X).

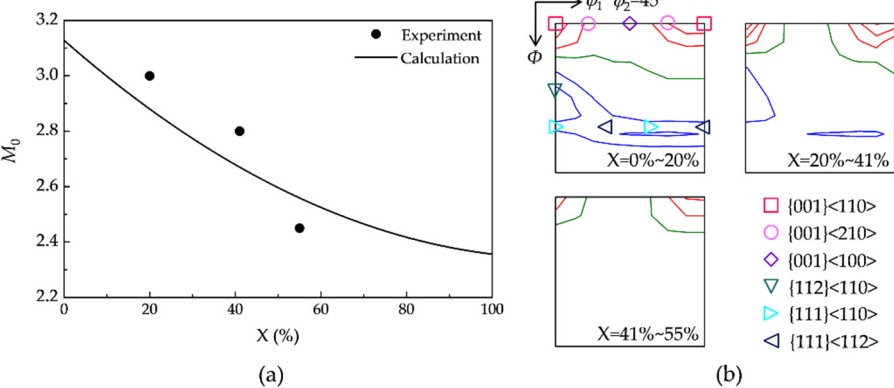

**Figure 7.** The calculated $M_0$ (**a**) and orientation density variation of main texture components (**b**) as a function of total low-Taylor-factor volume fraction increment (X). Red line levels: 1, 2, 3 . . . ; olive line level: 0; blue line levels: −1, −2, −3 . . . .

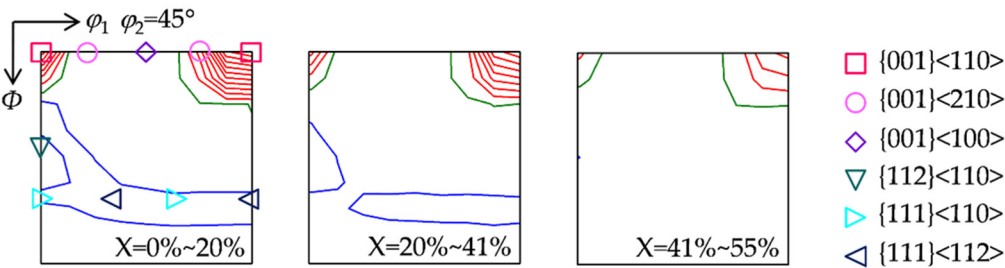

**Figure 8.** The calculated orientation density variation of main texture components as a function of total low-Taylor-factor volume fraction increment (X) with the model of Baczynski and Jonas. Red line levels: 1, 2, 3 . . . ; olive line level: 0; blue line levels: −1, −2, −3 . . . .

*3.5. Texture Evolution Rate*

According to Equations (3) and (4), $P_{g_i}^N$ and $\overline{M_{g_i}^A} - M_{g_i}$ or $M_{g_j} - \overline{M_{g_j}^A}$ can represent the texture evolution rate. Thus, the volume fraction increment rate of low-Taylor-factor orientation $g_i$ ($u_{g_i}$) is expressed as:

$$u_{g_i} = P_{g_i}^N \left( \overline{M_{g_i}^A} - M_{g_i} \right) \tag{7}$$

Similarly, the volume fraction decrement rate of high-Taylor-factor orientation $g_j$ ($u_{g_j}$) is written as:

$$u_{g_j} = M_{g_j} - \overline{M_{g_j}^A} \tag{8}$$

As shown in Figure 9, the nucleation probability ($P_{g_i}^N$) depends on the critical Taylor factor ($M_0$) and the investigated Taylor factor ($M$). With the decreasing $M_0$, $P_{g_i}^N$ and $u_{g_i}$ have a higher sensitivity to $M$ in the case of lower $M_0$, while $u_{g_j}$ exhibits a relatively lower sensitivity to $M$ regardless of $M_0$ range.

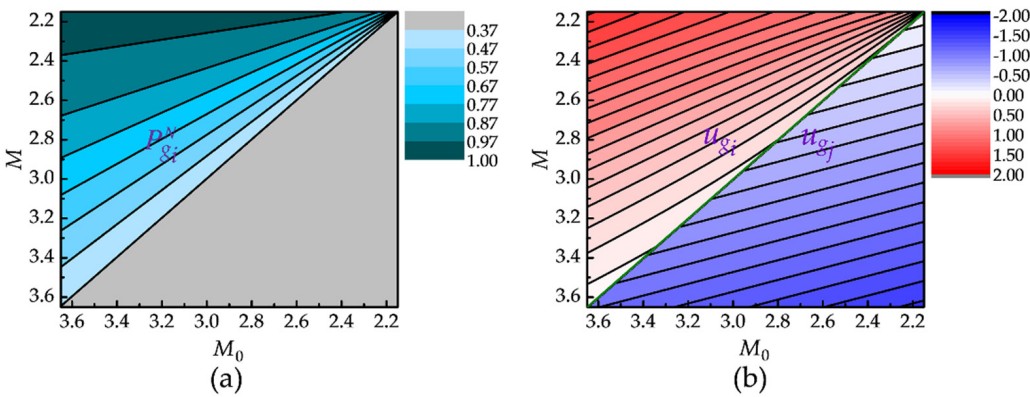

**Figure 9.** The contours of (**a**) nucleation probability ($P_{g_i}^N$), and (**b**) texture evolution rate ($u_{g_i}$ or $u_{g_j}$) dependent on critical Taylor factor ($M_0$). The olive line represents the boundary of low/high-Taylor-factor zones.

Under the present deformation conditions, {001}<110> and {001}<210> components lie in low-Taylor-factor zone, and {112}<110>, {111}<110> and {111}<112> components locate in high-Taylor-factor zone, while {001}<100> component situates around the olive line. The evolution rate of various texture components exhibits a distinct sensitivity to $M_0$. The increment rate of {001}<110> component is more sensitive to $M_0$ than {001}<210>, and the decrement rates of {112}<110>, {111}<110> and {111}<112> components are relatively less sensitive to $M_0$ than the increment rate of {001}<110> component.

Therefore, the present model can quantitatively describe the dynamic texture evolution by SIBM. The accurate prediction on texture development is highly valuable to design and control hot deformation texture, especially when low-Taylor-factor components are the expected target texture.

## 4. Conclusions

1. Strain-induced boundary migration occurs during high-temperature plane-strain compression of Fe-3.0 wt.% Si alloy, and various texture components have a distinct evolution with the critical Taylor factor changing continuously. The texture components with high Taylor factors are preferentially consumed, and the texture components with medium Taylor factors may experience a reversed change from enhancement to weakness with the proceeding strain-induced boundary migration.

2. Critical Taylor factor decreases continuously during the hot deformation process, and the evolution rate of various texture components has a distinct sensitivity to the

critical Taylor factor. With the decreasing critical Taylor factor, the increment rate of low-Taylor-factor orientation depends more sensitively on Taylor factor than the decrement rate of high-Taylor-factor orientation.

3. A quantitative model is proposed to describe texture evolution by incorporating the oriented nucleation probability dependent on a variable critical Taylor factor and the selective growth driven by a variable Taylor factor difference between adjacent grains. The model can efficiently capture the texture evolution by SIBM, as well as the sensitivity of critical Taylor factor, indicating the capability to predict and optimize hot deformation texture as a function of initial texture and dynamic process.

**Author Contributions:** Conceptualization, Y.S. and L.Z.; data curation, G.S.; formal analysis, G.S., X.C. and Y.S.; funding acquisition, Y.S., F.Z. and L.Z.; investigation, G.S.; Methodology, G.S., Y.S. and X.C.; project administration, Y.S., F.Z. and L.Z.; resources, Y.S. and F.Z.; supervision, Y.S.; validation, G.S., X.C. and Y.S.; Visualization, G.S., Y.S. and X.C.; writing—original draft preparation, G.S.; writing—review and editing, G.S., X.C., Z.H. and Y.S. All authors have read and agreed to the published version of the manuscript.

**Funding:** This work is financially supported by National Natural Science Foundation of China (Grant No. 51931002, 51671049).

**Institutional Review Board Statement:** Not applicable.

**Informed Consent Statement:** Not applicable.

**Data Availability Statement:** Data is contained within the article.

**Conflicts of Interest:** The authors declare that they have no conflict of interest in this work.

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
