# Peer review of "Texture Evolution by Strain-Induced Boundary Migration during Hot Deformation of Fe-3.0 wt.% Si Alloy: Experiment and Modeling"

_metals, doi:10.3390/met12020360_

Round 1

Reviewer 1 Report

 The article titled “Texture evolution by strain-induced boundary migration during hot deformation of Fe-3.0 wt.% Si alloy: Experiment and 3 modeling”

The article presents the texture analysis of the hot deformed Fe-3wt%Si alloy with some valuable results that deserve publication in Metal, however, the manuscript needs extensive revision and improvement. The following comments might help improves the article.

  1. Give some details about the experimental work conducted in the Abstract
  2. The introduction is weak and needs improvement.
  3. Why 15µm step size is used in all cases deformed and unreformed. Especially in the deformed, this might miss many deformation features.
  4. Section “3. Results” make it Results and Discussion” and do not repeat Results sections many times. Use subsections to present different results.
  5. Figure 2a : this is an “ inverse pole figure coloring map” or “image orientation map” use one of these instead of microstructure also indicates in the figure that the ODF section is calculated from this EBSD data.
  6. Section “3.1. Microstructure evolution during hot deformation” needs more description for the presented results. It will be much better if the grain size distribution is presented along with the OIM map.
  7. What is the objective of heating the specimens (Unreformed) for the same cycle of deformation? Comparison with the starting material can be enough
  8. Line 95 “Figure 3 shows the microstructures of undeformed and deformed parts of “ use “ inverse pole figure coloring map” or “image orientation map” for 
  9. The description given for Figure 3 is confusing as it does not refer to any notation (a, b, c …. etc), high resolution IPF maps are required to talk about fine grain and details of the grain boundaries.
  10. The description is given for Figure 4 also it does not refer to any notation (a, b, c …. etc),
  11. Define the abbreviation when first appeared “SIBM” inside the text.
  12. Section “5. Results“ It seems the authors mean conclusions here. Change it to conclusions.

The conclusions section needs more 

Author Response

Point 1: Give some details about the experimental work conducted in the Abstract.

Response 1: Thanks for the reviewer’s suggestion, details of the experimental work (lines 13-16 of page 1) are added in the Abstract.

Point 2: The introduction is weak and needs improvement.

Response 2: Thanks for the reviewer’s reminder, details of the importance of hot deformation texture for the needed texture during cold deformation and annealing have been added in the first paragraph of introduction (lines 34-42 of page 1).

Point 3: Why 15µm step size is used in all cases deformed and unreformed. Especially in the deformed, this might miss many deformation features.

Response 3: The average grain size in all deformed and undeformed parts of the specimens is larger than 150 µm. Thus, the step size of 15 µm of just 1/10 of the average grain size is appropriate to illustrate the general characterization and the feature of bulges can be clearly shown. Thanks for the reviewer’s reminder, a local enlarged region of Figure 3d with the step size of 4 µm (Figure 3i) is added to show the deformation features of small grains (lines 133-134 of page 4 and lines 149 of page 5).

Point 4: Section “3. Results” make it Results and Discussion” and do not repeat Results sections many times. Use subsections to present different results.

Response 4: Thanks for the reviewer’s reminder, the section “3. Results” has been changed to “3. Results and Discussion”, and the subsections to present different results have been presented.

Point 5: Figure 2a : this is an “ inverse pole figure coloring map” or “image orientation map” use one of these instead of microstructure also indicates in the figure that the ODF section is calculated from this EBSD data.

Response 5: Thanks for the reviewer’s suggestion, “microstructure” is changed to “orientation image map” (line 116 of page 3 and line 122 of page 4).

Point 6: Section “3.1. Microstructure evolution during hot deformation” needs more description for the presented results. It will be much better if the grain size distribution is presented along with the OIM map.

Response 6: Thanks for the reviewer’s suggestion, a local enlarged region of Figure 3d with 4 µm step to show the feature of small grains (Figure 3i) and the distribution of average grain size under different conditions of undeformed and deformed parts (Figure 3j) are added in section 3.1 (lines 133-134, 142-143 of page 4, line149 and lines 152-153 of page 5).

Point 7: What is the objective of heating the specimens (Undeformed) for the same cycle of deformation? Comparison with the starting material can be enough.

Response 7: The hot compression tests is carried out under various temperature and strain rate. This means that the undeformed zone actually experiences a heating process with different temperature and time. Thus, the comparison between undeformed and deformed parts of specimen is valuable to extract the effect of SIBM without the interference of normal grain growth.

Point 8: Line 95 “Figure 3 shows the microstructures of undeformed and deformed parts of “ use “ inverse pole figure coloring map” or “image orientation map” for 

Response 8: Thanks for the reviewer’s suggestion, “Figure 3 shows the microstructures of undeformed and deformed parts” is changed to “Figure 3 shows the orientation image maps of undeformed and deformed parts” (line 124 of page 3).

Point 9: The description given for Figure 3 is confusing as it does not refer to any notation (a, b, c …. etc), high resolution IPF maps are required to talk about fine grain and details of the grain boundaries.

Response 9: Thanks for the reviewer’s suggestion, the notation (a, b, c …. etc) has been addd in the description and high resolution IPF maps with 4 µm is added in Figure 3i to show the deformation features of small grains (lines 129, 133-134, 138, 140 and 147 of page 4).

Point 10: The description is given for Figure 4 also it does not refer to any notation (a, b, c …. etc).

Response 10: Thanks for the reviewer’s suggestion, the notation (a, b, c …. etc) has been addd in the description of Figure 4 (lines 157, 160, 161, 162 and 164 of page 5).

Point 11: Define the abbreviation when first appeared “SIBM” inside the text.

Response 11: The defination of abbreviation is presented in introduction part when first appeared “SIBM” (line 47 of page 2).

Point 12: Section “5. Results“ It seems the authors mean conclusions here. Change it to conclusions.

Response 12: Thanks for the reviewer’s reminder, “5. Results” is changed to “Conclusions”.

Point 13: The conclusions section needs more 

Response 13: Thanks for the reviewer’s suggestion, conclusions section has been added (lines 318-326 of page 10).

Reviewer 2 Report

The paper "Texture evolution by strain-induced boundary migration during hot deformation of Fe-3.0 wt.% Si alloy: Experiment and modeling" describes interesting experimental results about microstructure changes during the hot plane strain compression of the Si-containing electric steel. The paper is well written. The results were approved by the application of modern microstructural investigation techniques. However, the authors should highlight the scientific novelty of their work. Similar results were obtained previously by Onuki et. al [Ref 21]. The manuscript may be accepted for publication after revision accordingly following comments:

  1. The hot deformation process is not the last production stage for the electric steel. Cold rolling and annealing are usual procedures after the hot deformation to give the necessary magnetic properties. In the introduction part, the authors should describe in detail how the texture obtained during the hot deformation may influence the formation of the needed texture (usually Goss texture) during the cold deformation and annealing.
  2. The authors have investigated the hot deformation behaviour at two temperatures and different strain rates (0.01-1 1/s for 1150 ºC, and 5 1/s at 900 ºC). What was the reason for such different deformation conditions? The additional information should be added to the Materials and Methods part.
  3. The grain size obtained during the hot deformation is significantly influenced by the temperature and strain rate. However, as shown in Figure 3, there are no significant differences in the grain size of the samples, deformed at different strain rates (for the temperature of 1150 ºC) and heated to the different temperatures (Figure 3a, and c). The authors should describe such unusual behaviour, and provide the average grain size for the microstructures presented in Figure 3.
  4. Minor revisions are required:
  • The information about how the hot deformation temperature was controlled during the testing (especially, during the compression at high strain rates) should be added Materials and Methods part;
  • Most of the cited references are too old. The authors should consider more last references about the microstructure evolution of the steels during the hot deformation.

Author Response

Point 1: The hot deformation process is not the last production stage for the electric steel. Cold rolling and annealing are usual procedures after the hot deformation to give the necessary magnetic properties. In the introduction part, the authors should describe in detail how the texture obtained during the hot deformation may influence the formation of the needed texture (usually Goss texture) during the cold deformation and annealing.

Response 1: Thanks for the reviewer’s reminder, details of the importance of hot deformation texture for the needed texture during cold deformation and annealing has been added in the first paragraph of introduction (lines 34-42 of page 1).

Point 2: The authors have investigated the hot deformation behaviour at two temperatures and different strain rates (0.01-1 1/s for 1150 ºC, and 5 1/s at 900 ºC). What was the reason for such different deformation conditions? The additional information should be added to the Materials and Methods part.

Response 2: Thanks for the the reviewer’s suggestion, the deformed specimens during hot compression are required to have different states. One is the deformed state without SIBM, indicating the sole effect of orientation rotation by plastic deformation. The others are the deformed states with different progresses of SIBM, which results from the combination of orientation rotation by plastic deformation and bulges by SIBM. The initial materials of Fe-3.0 wt.% Si specimens prepared for plane-strain compression are characterized with equiaxed grains, which are annealed by 1150 °C×10 min. According to the stress-strain curve, the strain of 50% of the deformation conditions (the strain rates of 1, 0.1, and 0.01 s−1 at 1150 °C as well as 5 s−1 at 900 °C) is in the steady stage. Thus, after deformation at 900 °C with 5 s−1, most grains are highly elongated along RD direction, and the microstructure can be deemed as the deformed state without SIBM. With increasing deformation temperature and decreasing strain rate, the effect of SIBM enhances. Thus, after deformation at 1150 °C with 1, 0.1, and 0.01 s−1, many grain boundaries are featured with different bulge amplitudes, and the microstructure can be considered as the deformed states with different progresses of SIBM. Accordingly, detail descriptions have been added in materials and method sections (lines 93-95 of page 2).

Point 3: The grain size obtained during the hot deformation is significantly influenced by the temperature and strain rate. However, as shown in Figure 3, there are no significant differences in the grain size of the samples, deformed at different strain rates (for the temperature of 1150 ºC) and heated to the different temperatures (Figure 3a, and c). The authors should describe such unusual behaviour, and provide the average grain size for the microstructures presented in Figure 3.

Response 3: Thanks for the the reviewer’s suggestion, Figures 3a, c, e and g show the microstructures of undeformed parts, which experience the same heating history as deformed parts under various deformation temperature and strain rate. The specimens were annealed at 1150 °C for 10 min prior to hot compression, while the heating times during the compression at 1150 ºC with 1, 0.1, and 0.01 s−1 are 60 s, 65s and 110s respectively, so there are no significant differences in the grain size of undeformed parts. However, the grain size of deformed parts is significantly influenced by the temperature and strain rate, as shown in Figures 3b, d, f and h. Accordingly, the average grain size for the microstructures of undeformed and deformed parts of the compressed specimens under different deformation conditions is presented in Figure 3j (lines 142-133 of page 4 and line 149, 153 of page 5).

Point 4: The information about how the hot deformation temperature was controlled during the testing (especially, during the compression at high strain rates) should be added Materials and Methods part.

Response 4: Thanks for the the reviewer’s suggestion, the information about the control of deformation temperature is added in Materials and Methods part (lines 89-91 of page 2).

Point 5: Most of the cited references are too old. The authors should consider more last references about the microstructure evolution of the steels during the hot deformation.

Response 5: Thanks for the the reviewer’s suggestion, the cited references has been renewed, and several latest articles about the microstructure evolution of the steels are added (References [12-16]).

Reviewer 3 Report

The abstract requires some brief quantitative details

Please avoid block citation such as 3-10, 11-18 and so on discuss each separately . this apply for the entire paper

“the detailed texture evolution throughout” I think you should refer to this specific alloy cause you are not able to refer to all !

Some critical review is required now it is very briefly and not much details why the literature actually do not respond to research questions

“10 °C·s−1 to the target deformation” seems quite slow heating rate for industrial application for example

Please provide details of sample preparation for EBSD cause now is just some last step but other are required too

You said “15 mm×20 mm” but for example figure 2 a is much smaller !

“The specimens are completely recrystallized with average grain size of 150 μm” from your figures seems that yet are grain much bigger and smaller one – otherwise please provide details how do you quantify actually the recrystallization completion ?

Line 95-96 are little big vague as the ebsd is not reconstructed !

How do you exactly define this “highly elongated with small bulges of original grain boundaries” for example how do you define elongation ?

For example line 129-131 requires some citation in order to prove it

 A section of proper discussion is required

Line 260 may be is conclusion rather the results

The conclusions are too superficial – quantitative details are required

The state of art presented is quite out of date

Author Response

Point 1: The abstract requires some brief quantitative details.

Response 1: Thanks for the the reviewer’s suggestion, brief quantitative details (lines 17-21 of page 1) have been added in the abstract.

Point 2: Please avoid block citation such as 3-10, 11-18 and so on discuss each separately. this apply for the entire paper

Response 2: Thanks for the reviewer’s suggestion, block citation has been changed.

Point 3: “the detailed texture evolution throughout” I think you should refer to this specific alloy cause you are not able to refer to all !

Response 3: Thanks for the reviewer’s reminder, the sentence “the detailed texture evolution throughout hot deformation has not yet been clearly clarified” has been revised more specifically into “the texture evolution process during hot deformation assisted by SIBM has not yet well been described in consideration of a changing Taylor factor range” (lines 73-75 of page 2).

Point 4: “10 °C·s−1 to the target deformation” seems quite slow heating rate for industrial application for example

Response 4: Thanks for the reviewer’s reminder, “10 °C·s−1 to the target deformation” is really a slow heating rate for industrial application, however, it has no significant effect on the microstructure prior to hot compression because the hot rolled plates used for plane-strain compression are annealed at 1150 °C for 10 min to complete recrystallization.

Point 5: Please provide details of sample preparation for EBSD cause now is just some last step but other are required too.

Response 5: Thanks for the reviewer’s suggestion, details of sample preparation for EBSD are added (lines 104-107 of page 3).

Point 6: You said “15 mm×20 mm” but for example figure 2 a is much smaller !

Response 6: Thanks for the reviewer’s reminder, to facilitate the EBSD measurement, a scanned area of “5 mm×20 mm” for one specimen is cut into two parts with the area of “5 mm×10 mm”. Three specimens, a total area of about 15 mm×20 mm, are scanned for each deformation condition. Accordingly, Figure 2a is a typical area to illustrate the microstructure with the area of about “5 mm×10 mm”, and details of the scanned area are added in Materials and Methods part (lines 109-111 of page 3).

Point 7: “The specimens are completely recrystallized with average grain size of 150 μm” from your figures seems that yet are grain much bigger and smaller one – otherwise please provide details how do you quantify actually the recrystallization completion ?

Response 7: Thanks for the reviewer’s reminder, the distribution of grain size is not uniform which consists much larger and smaller ones. Thus, the sentence has been revised into “The initial microstructure mainly consists of equiaxed grains distributed between 100~400 μm” (lines 117-118 of page 3).

Point 8: Line 95-96 are little big vague as the ebsd is not reconstructed !

Response 8: Thanks for the review’s reminder, Figure 2 illustrates the orientation image map and texture of hot compression specimens reconstructed from EBSD data (lines 116-117 of page 3).

Point 9: How do you exactly define this “highly elongated with small bulges of original grain boundaries” for example how do you define elongation ?

Response 9: The highly elongated grains in morphology are defined by the aspect ratio of grain significantly larger than 1. The initial grains prior to hot compression are nearly equiaxed with the aspect ratio close to one. During deformation, the aspect ratio of grain is changing. If the aspect ratio of the grains is far from one, the boundaries are considered to be elongated.

Point 10: For example line 129-131 requires some citation in order to prove it.

Response 10: Thanks for the review’s reminder, the citation has been added (lines 164-167 of page 6).

Point 11:  A section of proper discussion is required.

Response 11: Thanks for the reviewer’s suggestion, a section of discussion is added (lines 271-286 of page 9).

Point 12: Line 260 may be is conclusion rather the results.

Response 12: Thanks for the review’s reminder, “results” has changed to “conclusions” (line 315 of page 10).

Point 13: The conclusions are too superficial – quantitative details are required.

Response 13: Thanks for the reviewer’s suggestion, quantitative details have been added in section of conclusions (lines 318-326 of page 10).

Round 2

Reviewer 1 Report

The manuscript has been improved after careful consideration of the comments, however, a few amendments need to be carried out before publication.

  The abstract needs to be supported with more results in terms of texture components and grain size for example.

“ Nucleation and growth assisted by strain-induced boundary migration result in strong texture components with low Taylor factors.” Give the main texture components here.

“The specimens for EBSD analysis were 112 prepared by first mechanical polishing and then electropolishing in a solution of 92% ethanol and 8% perchloric acid for 15 s at 20 V.” Mention here the temperature at which the electropolishing was conducted

Author Response

Point 1: The abstract needs to be supported with more results in terms of texture components and grain size for example.

Response 1: Thanks for the reviewer’s suggestion, results in terms of texture components and grain size (lines 17-19 of page 1) are added in the Abstract.

Point 2: “ Nucleation and growth assisted by strain-induced boundary migration result in strong texture components with low Taylor factors.” Give the main texture components here.

Response 2: Thanks for the reviewer’s suggestion, the main texture components are added (line 17 of page 1).

Point 3: “The specimens for EBSD analysis were prepared by first mechanical polishing and then electropolishing in a solution of 92% ethanol and 8% perchloric acid for 15 s at 20 V.” Mention here the temperature at which the electropolishing was conducted

Response 3: Thanks for the reviewer’s reminder, the temperature of electropolishing is added (line 114-115 of page 3).

Reviewer 2 Report

The authors have answered the previous comments and made necessary changes to the manuscript. The paper may be accepted after minor correction:

1. The standard deviations should be added to the average values of the grain size in Figure 3j.

Author Response

Point 1: The standard deviations should be added to the average values of the grain size in Figure 3j.

Response 1: Thanks for the reviewer’s reminder, the standard deviations are added to the average values of the grain size in Figure 3j.

Reviewer 3 Report

My comments were well addressed therefore I suggest acceptance.

Author Response

Thanks for your comments and suggestions, they are very helpful for improving our paper.